# Finding the Best Match: A Ranking Procedure of Fish Species Combinations for Polyculture Development

**DOI:** 10.3390/life12091315

**Published:** 2022-08-26

**Authors:** Nellya Amoussou, Marielle Thomas, Alain Pasquet, Thomas Lecocq

**Affiliations:** 1Unit Research Animal and Functionality of Animal Products (UR AFPA), University of Lorraine (UL), Institut National de la Recherche pour l’Agriculture, l’Alimentation et l’Environnement (INRAE), 54000 Nancy, France; 2Zone Atelier Moselle (LTER), 54506 Vandœuvre-lès-Nancy, France

**Keywords:** polyculture, stakeholder priorities, multivariate analysis, recirculated aquaculture systems

## Abstract

Polyculture is a potentially interesting rearing practice for future aquaculture developments. Nevertheless, it may result in beneficial as well as detrimental consequences for fish production. One way to maximize the benefits of polyculture is to combine species with high levels of compatibility and complementarity. This requires the development of a ranking procedure, based on a multi-trait assessment, that highlights the most suitable species combinations for polyculture. Moreover, in order to ensure the relevance of such a procedure, it is important to integrate the socio-economic expectations by assigning relative weights to each trait according to the stakeholder priorities. Here, we proposed a ranking procedure of candidate fish polycultures (i.e., species combinations that could be potentially interesting for aquaculture) based on a multi-trait assessment approach and the stakeholder priorities. This procedure aims at successively (i) weighting evaluation results obtained for each candidate polyculture according to stakeholder priorities; (ii) assessing differentiation between candidate species combinations based on these weighted results; and (iii) ranking differentiated candidate polycultures. We applied our procedure on three test cases of fish polycultures in recirculated aquaculture systems. These test cases each focused on a target species (two on *Sander lucioperca* and one on *Carassius auratus*), which were reared in two or three different alternative candidate fish polycultures. For each test case, our procedure aimed at ranking alternative combinations according to their benefits for production and/or welfare of the target species. These benefits were evaluated based on survival rate as well as morphology, behavioral, and physiological traits. Three scenarios of stakeholder priorities were considered for weighting evaluation results: placing a premium on production, welfare, or both for the target species. A comparison of our procedure results between these scenarios showed that the ranking changed for candidate polycultures in two test cases. This highlights the need to carefully consider stakeholder priorities when choosing fish polycultures.

## 1. Introduction

In the coming decades, the human demand for food products is expected to double, due to world population growth [1]. Human nutrition partially relies on aquatic organisms [2]. Until the early 2010′s, the global supply of aquatic products was mainly provided by fisheries [3]. However, in recent years, aquaculture has become an increasingly important sector for this supply because wild fisheries are no longer able to meet the demand for aquatic products [3]. In Western countries, intensive monoculture (i.e., production of a single species in cages, recirculated aquaculture systems [RASs], or ponds) has been favored in aquaculture development [4]. However, monoculture may have several limitations. First, it can limit the food conversion ratio [5], which compromises the water quality of the system by increasing nutrient concentration (generated by non-digested feed) in the effluents [6]. Second, monoculture has also been criticized from human food security and economic viewpoints, as it might have a low resilience [7] and a low adaptative potential to face changes in environmental and socio-economic contexts [8]. Third, there is now a strong societal expectation to ensure animal welfare, which may be negatively impacted by intensive production [9].

Fish polyculture is the production of two or more species in the same physical space at the same time [10]. Recent studies have revived interest in this ancient rearing practice because it may improve current aquaculture developments and partially overcome fish-intensive monoculture limitations [5,11]. Indeed, polyculture can enhance farming efficiency by improving the use of resources that are naturally present or added in the fish-rearing environment [12] and by recycling nutrients from farmed biomass [13,14] in extensive and intensive aquaculture [15]. This can decrease environmental impacts and improve the sustainability of production systems [8,16,17], and it can even increase the fish production [18]. Nevertheless, polycultures are complex aquaculture systems because co-farming can impact the survival [19,20], biological functions (e.g., nutrition [21] and relationships [22,23] and thus production performances of the farmed species [15]). Moreover, it can also result in animal welfare issues [15]. Actually, polyculture can be beneficial for rearing, provided that species compatibility (i.e., ability of species to live in the same production system while minimizing detrimental interactions or competition for trophic or spatial available resources) and, even more, complementarity (i.e., co-farmed species can use different portions of resources or display commensal/mutualistic interactions) occur between the combined species [15]. Therefore, one way to maximize the benefits of polyculture is to integrate fish species diversity by comparing several possible fish combinations (i.e., candidate polycultures) for a particular purpose (i.e., developing polyculture around a target species and/or in a particular rearing system), and eventually, highlighting the combination(s) with the highest level of compatibility and/or complementarity [15]. This places a premium on a standardized assessment procedure to rank fish species combinations in order to select valuable polycultures.

Such a procedure must be built by considering four important aspects. First, since polyculture can impact many components of fish biology and aquaculture, assessment of candidate fish species combinations must be based on traits related to different biological functions, reflecting species compatibility and beyond those involved in socio-economical concerns [15]. Overall, these traits include features related to zootechnical performances, such as morphology and growth traits [24,25,26], and responses to stress, such as behavioral [27] and physiological traits [28]. Second, each of these traits can be polarized according to the sought expression of the trait for aquaculture purposes (e.g., highest growth rate and lowest stress response). These sought expressions must be integrated into the multi-trait framework. Third, the polarized traits are not all equally important. Thus, they can be weighed in the multi-trait framework based on their relative importance in the priorities (e.g., production and/or welfare [29,30,31]) of stakeholders (i.e., fish farmers, consumers, scholars, policy makers, and non-governmental organizations). The importance of integrating these priorities in decision-making approaches for aquaculture development has already been underlined in several studies e.g., [32,33]. Fourth, it is worth noting that multi-trait assessment may provide conflicting results between equally important traits (e.g., a polyculture can have a positive impact on growth but negative consequences on fish welfare). This can complicate decision-making for production development (see such an issue in the multi-trait evaluation of monoculture in [33]). This requires making a synthesis of potentially conflicting results by integrating them into a score to facilitate the ranking (see similar solution in [34]).

To our knowledge, no standardized assessment procedure, based on stakeholder priorities to rank candidate polycultures, has been proposed to date. Therefore, we propose such a procedure for fish polycultures, which is a multi-trait assessment based on four steps: (i) the selection of biological traits to assess candidate polycultures; (ii) the dataset treatment (e.g., polarization); (iii) the consideration of stakeholder priorities in the assessment by a weighting approach; and (iv) the integration of the results with a multivariate analysis and a ranking of the candidate polycultures. We applied our ranking procedure on three test cases of fish polycultures in RAS.

## 2. Materials and Methods

### 2.1. A Four-Step of the Ranking Procedure

Since polyculture is considered a potentially more advantageous solution compared to monoculture, we developed a procedure to assess candidate polycultures that has been thought of as a way to improve the rearing of a target species by combining it with other species. Thus, the monoculture of the target species was used as the baseline information to quantify the potential benefits of polyculture compared to traditional monoculture.

#### 2.1.1. The First Step: Biological Trait Selection

In order to assess and rank candidate polycultures, the procedure considered several categories of traits related to (i) survival and morphology and (ii) behavior and physiology.

Survival and morphological traits

Producing a lot of fish with good growth is of paramount importance in aquaculture. Therefore, measuring the survival rate (SR), the final weight (Wf), and the specific growth rate (SGR) is an important insight to assess fish production [5,11,35]. The disparity in growth among individuals (i.e., size variation within a stock of fish specimens with similar age) is another piece of important information for aquaculture. Indeed, it is seen as a proxy of competition occurring among individuals for food; the more restricted or defensible the food supply, the greater the competition and the more pronounced the disparity in food acquisition among individuals [36,37]. It can be assessed by examining the weight heterogeneity (CV) [37] between fish species. The Fulton condition index (FCF) is also an important piece of information for aquaculture, as it reflects the potential fitness for fish [38]. The fitness of a fish is an important insight to assess the welfare of fish: a decline of FCF is directly equivalent to a decrease in the energetic reserves of fish [39].

2.Behavioral and physiological traits

Nowadays, it is of utmost importance to integrate animal welfare concerns in animal production [40,41]. Traits related to stress responses are regarded as relevant proxies for welfare assessment. Such traits include behavioral (i.e., agonistic [Agr] and flight [Flg]) and physiological traits (i.e., hematocrit [Hct], glucose [Glu], cortisol [Cort], serotonin [Ser], and dopamine [Dop]). While Hct, Glu, Cort, Ser, and Dop are considered key markers of the physiological responses of fish, Agr and Flg can act as indicators of competition between species [23,42,43,44] and thus identify potential compatibility issues.

#### 2.1.2. The Second Step: Dataset Treatment

First, we computed the “delta value” (ΔVal) for all pair-wises between the replicates of each polyculture and the replicates of the monoculture. For each pair-wise, we subtracted the values of traits from each monoculture from those of polyculture. This allowed the highlighting of the potential benefits of polyculture compared to the monoculture of the target species. Second, each ΔVal was polarized according to the sought expression of the corresponding trait for the aquaculture purpose (i.e., multiplying by +1 the ΔVal related to the traits for which a higher expression level was sought and by −1 those for which a lower expression level was sought; see Table 1 for more information). Third, we standardized the ΔVal. Standardization is a technique for comparing data that are measured with different units (e.g., SR is expressed in “%”, Wf and Cort were respectively expressed in “g” and “ng.mL^−1^”). The standardized values corresponded to the centered scaled value (i.e., ranged between 0 and 1: R-package [scales]) and were attributed to the ΔVal for each trait. Fourth, we selected relevant and the least-correlated ΔVal of the traits (R^2^ < 0.9) by means of Ward D2 linkage cluster analysis (cor: method = Pearson, from R-package ape [45] in R [46]).

#### 2.1.3. The Third Step: Weighting the Traits

To exemplify the consequences of different stakeholder priorities, we applied three alternative frameworks (arbitrarily defined) of weighting coefficients (Wc) on the ΔVal corresponding to three types of priorities: (i) one in which all trait categories are regarded as equally important, called “neutral-weighting;” (ii) one in which priority is given to the SR and morphological traits (except the FCF), called “production-weighting;” or (iii) one in which priority is given to SR, FCF, behavioral, and physiological traits, called “welfare-weighting.” According to “neutral-weighting,” all ΔVals were multiplied by Wc = 1. When assuming “production-weighting,” the ΔVals related to SR, CV, Wf, and SGR were multiplied, respectively, by Wc > 1 (three levels of weighting were applied in three independent assessments: 2, 5 or 10), while the others remained unchanged (Wc = 1). When assuming “welfare-weighting,” the ΔVals related to SR, FCF, Agr, Flg, Hct, Glu, Cort, Ser, and Dop were multiplied, respectively, by Wc > 1 (2, 5, or 10), while the others remained unchanged (Wc = 1). See Table 2 for more information.

#### 2.1.4. The Fourth Step: Integrating the Results

We highlighted significant differences between the polycultures (*p*-value < 0.05) through a multivariate permutation test (MRPP: Euclidean distance matrix method; permutations = 10,000; R-package vegan [47]; R software [46]). Global MRPP was applied first, and then, when a significant differentiation between candidate polycultures was returned, the pair-wise comparison MRPPs with a Bonferroni correction were performed. We used a MRPP because parametric test assumptions for the normality of a multivariate dataset (mshapiro.test from R-package mvnormtest [48]; R software [46]) were not met. When a candidate polyculture was significantly divergent from other polycultures, we calculated the weighted sum of its ΔVals (*ws*: Formula (1)), the rank of the *ws*, and the sum of the ranks (*rs*), and divided by the number of replicates of the candidate polycultures. The rank of each polyculture was defined through the *rs* values. The candidate polyculture with the lowest *rs* was considered the best candidate for polyculture.
(1)$$ws=\overset{i=n}{\underset{i=1}{\sum }}\left[\frac{\Delta Val\;\left(i\right)}{nR}\times \mathrm{Wc}\right]$$
where *n*, *nR*, Δ*Val*, and Wc corresponded, respectively, to the number of considered traits, the number of replicates of the candidate polycultures, the delta value, and the weighting coefficients. (That could correspond to 1, 2, 5, or 10, according to the priorities.)

### 2.2. Test Cases

As test cases for our procedure, we used three datasets of traits obtained from the polyculture trials (see Table 3 and [49] for more information about the trials), and we analyzed them [50,51]).

For each test case, all species combinations were centered on a target species (i.e., pikeperch [*Sander lucioperca*] or goldfish [*Carassius auratus*]) (currently produced in monoculture) for which we aimed at improving the production and welfare through polyculture. Pikeperch and goldfish were chosen as target species for their high economic importance: pikeperch is an increasingly important species for European aquaculture [52] and goldfish is an ornamental species highly requested by the aquariology industry [53,54]. These target species were combined with other species that are also commercially interesting, either for a large (common carp [*Cyprinus carpio*], black-bass [*Micropterus salmoides*], and sterlet [*Acipenser ruthenus*]) or niche market (European-perch [*Perca fluviatilis*] and tench [*Tinca tinca*]]), or of a potential economic interest (ruffe [*Gymnocephalus cernua*]) [55,56,57,58,59,60,61,62].

## 3. Results

### 3.1. The Test Case 1

**Step 1**. 12 traits (Table 3) were considered. **Step 2**. Since we detected a strong correlation between the ΔVals of Wf and SGR (R^2^ = 1, *p*-value < 0.01) and the ΔVals of Ser and Dop (R^2^ = 0.99, *p*-value < 0.01), the ΔVals of SGR and Dop were retained for the analysis. (A random selection between correlated ΔVal of traits was applied to select which one was retained.) Finally, the ΔVals of 10 traits were used for the analysis (for more information, see [63]). **Step 3**. The ΔVals were kept unchanged with the neutral-weighting. For the production-weighting, the ΔVals of SR, CV, and SGR were multiplied by Wc > 1 and the ΔVals of the other traits were not changed (Wc = 1). For the welfare-weighting, the ΔVals of SR, FCF, Hct, Glu, Cort, Dop, Agr, and Flg were multiplied by Wc > 1, and the ΔVals of the other traits were not changed (Wc = 1) (see Table 2 for more information). **Step 4**. For all weightings, a global MRPP and the pair-wise MRPP (see Table 4A,B) showed significant differences between polycultures (*p*-value < 0.05). The ranking of candidate polycultures was then performed. This ranking changed according to the types of priority and relative weight put on them (Table 4C).

### 3.2. The Test Case 2

**Step 1**. 10 traits (Table 3) were considered. **Step 2**. Since we detected a strong correlation between the ΔVals of SGR and FCF (R^2^ = 1, *p*-value < 0.01) and the ΔVals of Glu and Cort (R^2^ = 1, *p*-value < 0.01), the ΔVals of SGR and Cort were retained for the analysis. (A random selection between correlated ΔVals of traits was applied to select which one was retained.) Finally, the ΔVals of 8 traits were used for the analysis (for more information, see [63]). **Step 3**. The ΔVals were kept unchanged with the neutral-weighting. For the production-weighting, the ΔVals of SR, Wf, and CV were multiplied by Wc > 1, and the ΔVals of the other traits were not changed (Wc = 1). For the welfare-weighting, the ΔVals of SR, FCF, Hct, Cort, Agr, and Flg were multiplied by Wc > 1, and the ΔVals of the other traits were not changed (Wc = 1) (see Table 2 for more information). **Step 4**. For all weightings, a global MRPP (see Table 5A) showed significant differences between polycultures (*p*-value < 0.05). The ranking of candidate polycultures was then performed. This ranking did not change, according to the types of priority and relative weight put on them (Table 4B).

### 3.3. The Test Case 3

**Step 1**. Seven traits (Table 3) were considered. **Step 2**. No correlation was detected between the ΔVals of the traits (for more information, see [63]). Consequently, all ΔVals of the seven traits were used for the analysis. **Step 3**. The ΔVals were kept unchanged with the neutral-weighting. For the production-weighting, the ΔVals of SR, CV, Wf, and SGR were multiplied by Wc > 1, and the ΔVals of other traits were not changed (Wc = 1). For the welfare-weighting, the ΔVals of SR, FCF, Ser, and Agr were multiplied by Wc > 1, and the ΔVals of other traits were not changed (Wc = 1) (see Table 2 for more information). **Step 4**. For all weightings, a global MRPP and the pair-wise MRPP (see Table 6A,B) showed significant differences between polycultures (*p*-value < 0.05). The ranking of candidate polycultures was then performed. This ranking changed according to the types of priority and relative weight put on them (Table 6C).

## 4. Discussion

### 4.1. The Importance of Considering Stakeholder Priorites to Rank Polycultures

Our procedure allows the ranking of candidate polycultures for the development of aquaculture. In our test cases, it is worth noting that the ranking changed according to the applied weighting, reflecting different types of stakeholder priorities (Table 4, Table 5 and Table 6). Such an impact of stakeholder priorities on the ranking/decision-making process has already been highlighted in aquaculture (e.g., to find best candidate species or population for aquaculture [34,64]), as well as in terrestrial production [65,66,67] or applied ecology (e.g., adaptation strategy to climate change [68]). In our analyses, the ranks of some candidate polycultures also changed, depending on the Wc (2, 5, or 10) in two test cases (see a similar observation in [69] for renewable energy sources).

This raises the question of the relevance of advising a given polyculture over another when the ranking seems to be not very robust. The user of the procedure must then consider the result of the evaluation with caution and consider other criteria (economic value or availability of species) to decide between two candidate polycultures whose rankings are swapped, according to the applied Wc. Conversely, when the ranking of a polyculture remains the same, regardless of the Wc applied on the delta values (the test case 2), the user may consider having a robust result to choose from or to discard a candidate polyculture over the others.

### 4.2. Ranking Procedure Limitations

In the procedure, users should be aware of three main limitations.

First, the ranking of polycultures is only relevant for future applications if it is applied in a farming context similar to the experimental conditions used to establish this ranking. This environmental dependence is due to the fact that the physico-chemical characteristics, biomass, densities, or ratios among fish species can potentially impact fish performances and behavior and thus polyculture consequences [70,71,72].

Second, the procedure can also be challenging because the measurement of some traits can be hard to achieve in some rearing environments. For instance, while morphological traits are easily measurable on fish in most rearing systems [11,73], the measurement of behavioral traits (i.e., interspecific interactions) depends on the technics and parameters, which are more complex to set up and obtain. Indeed, the collection of interspecific interaction data requires an experimental facility, favorable to the observation [11,23], which are more difficult to obtain, for instance, in a pond. This means the list of relevant traits might need to be adapted for some fish-rearing systems.

Third, this procedure can be viewed as heavy-going, since it is based on a multi-trait assessment, which requires costly and potentially time-consuming experiments and analyses (e.g., behavioral and physiological traits) and potentially invasive measurements (e.g., physiological traits). One solution to facilitate the multi-trait assessment is to use traits or proxies of relevant traits that are easy to measure to increase the feasibility of the procedure (see similar rationalization for another multi-trait assessment in [74]).

### 4.3. What Is Next?

We argue that the current version of the ranking procedure is already usable for aquaculture purposes, but further improvements are needed to ensure its efficiency and operationality for future users.

First, our ranking procedure focused on a target species (i.e., pikeperch and goldfish), disregarding the polyculture consequences for all the associated fish species. However, polyculture can be beneficial for some species at the detriment of others [22,75]. This highlights the need for a global ranking (i.e., considering the monocultures for the associated fish species for each candidate polyculture). The global ranking might be considered a mean of the *ws* (i.e., sum of *ws* of the traits related to each combined species divided by the number of combined species). The future users of the global ranking should be aware that its application allows for finding the best polyculture among those compared by considering all the combined species, but it does not exclude the possibility that all the tested polycultures can lead to negative consequences on all or some of the combined species compared to the reference monocultures.

Second, in this study, the Wcs have been arbitrarily assigned to the ΔVals of the traits, since we aimed to demonstrate that, depending on the priorities of the stakeholder, this coefficient could modify the rank of the polycultures. However, this procedure must be applied through a co-construction with stakeholders, by involving them using a survey e.g., [76,77]. Indeed, stakeholder involvement in the development stages of aquaculture is particularly useful [33,78]. This allows the defining of a relevant Wc for a particular polyculture purpose. Moreover, stakeholder inclusion in the ranking procedure means social acceptability [79,80,81]. The co-construction with stakeholders could also provide an opportunity to extend our procedure to other factors (i.e., economic considerations) in order to increase the relevance of the resulting rankings.

## Figures and Tables

**Table 1 life-12-01315-t001:** Polarization of delta values of traits (SR = survival rate, Wf = final weight, SGR = specific growth rate, CV = coefficient of variation, FCF = Fulton condition factor, Agr = agonistic, Flg = flight, Hct = hematocrit, Glu = glucose, Cort = cortisol, Ser = serotonin and Dop = dopamine). The polarization considered the sought expression in aquaculture for each trait.

Categories	Traits	Polarized Delta Values (ΔVal)
Survival	SR	+SR
Morphological	Wf	+Wf
	SGR	+SGR
	CV	−CV
	FCF	+FCF
Behavioral	Agr	−Agr
	Flg	−Flg
Physiological	Hct	−Hct
	Glu	−Glu
	Cort	−Cort
	Ser	−Ser
	Dop	−Dop

**Table 2 life-12-01315-t002:** Application of the weighting coefficients (Wc) on each delta value (ΔVal), according to the traits considered in this study (SR = survival rate, Wf = final weight, SGR = specific growth rate, CV = coefficient of variation, FCF = Fulton condition factor, Agr = agonistic, Flg = flight, Hct = hematocrit, Glu = glucose, Cort = cortisol, Ser = serotonin, and Dop = dopamine).

	Delta Values (ΔVal)											
Priorities	SR	Wf	SGR	CV	FCF	Agr	Flg	Hct	Glu	Cort	Ser	Dop
Neutral-weighting	1	1	1	1	1	1	1	1	1	1	1	1
Production-weighting	2, 5 or 10	2, 5 or 10	2, 5 or 10	2, 5 or 10	1	1	1	1	1	1	1	1
Welfare-weighting	2, 5 or 10	1	1	1	2, 5 or 10	2, 5 or 10	2, 5 or 10	2, 5 or 10	2, 5 or 10	2, 5 or 10	2, 5 or 10	2, 5 or 10

**Table 3 life-12-01315-t003:** Main information related to the three test cases in recirculated aquaculture systems. “Polycultures” shows compared fish combinations for each test case; underlined species are the target species of each test case. “Rearing volume” displays the volume of the experimental system in which the test cases were studied. “Trial duration” provides the duration of the experiment for each test case. “Traits” lists studied traits for each test case (SR = survival rate, Wf = final weight, SGR = specific growth rate, CV = coefficient of variation, FCF = Fulton condition factor, Agr = agonistic, Flg = flight, Hct = hematocrit, Glu = glucose, Cort = cortisol, Ser = serotonin, and Dop = dopamine); for test cases 1 and 3, all traits were evaluated based on three replicates, except for those marked with *, for which two replicates were used; for test case 2, all traits were evaluated based on five replicates, except for those marked with **, for which four replicates were used. “Measurement time” shows when or over which time period the traits were measured.

Test Cases	Polycultures	Rearing Volume (m^3^)	Trial Duration (Days)	Traits	Measurement Time (Days)
1	Pikeperch, common carp [**SC**]	2	90	SR, SGRWf, CV, FCF, Hct, Glu, Cort, Ser, DopAgr *, Flg *	Between 0–90At 90At 48

Pikeperch, black-bass [**SM**]

Pikeperch, common carp, European-perch [**SCP**]

2	Goldfish, roach, ruffe [**CRG**]	0.3	90	SR, SGRWf, CV, FCF, Hct, Glu, CortAgr **, Flg **	Between 0–90At 90At 48

Goldfish, roach, European-perch [**CRP**]

3	Pikeperch, tench [**PT**]	0.3	60	SR, SGRWf, CV, FCF, SerAgr *	Between 0–60At 60At 32

Pikeperch, sterlet [**PS**]

Pikeperch, sterlet, tench [**PST**]


**Table 4 life-12-01315-t004:** (A) Global, (B) Pair-wise results of multivariate permutation tests and (C) ranking results. The candidate polycultures corresponded to SM (pikeperch, black-bass), SC (pikeperch, common carp), and SCP (pikeperch, common-carp, European-perch). *rs* corresponds to the sum of the ranks. The numbers in brackets corresponds to the level of weighting. Significance = *p* < 0.05.

**(A)**											
**Priorities**				**Weighting Coefficients**							
				1		2		5		10	
Neutral-weighting				df = 2		-		-		-	
				A = 0.197							
Production-weighting				-		df = 2		df = 2		df = 2	
						A = 0.324		A = 0.381		A = 0.396	
Welfare-weighting				-		df = 2		df = 2		df = 2	
						A = 0.213		A = 0.199		A = 0.197	
**(B)**											
**Priorities**		**Polycultures**				**SC**		**SM**			
Neutral-weighting		**SM**				df = 1					
						A = 0.224					
		**SCP**				df = 1		df = 1			
						A = 0.197		A = 0.198			
Production-weighting		**SM**				df = 1					
						A = 0.245[2], 0.273[5], 0.282[10]					
		**SCP**				df = 1		df = 1			
						A = 0.250[2], 0.325[5], 0.350[10],		A = 0.316[2], 0.424[5], 0.452[10]			
Neutral-weighting		**SM**				df = 1					
						A = 0.222[2], 0.222[5], 0.222[10]					
		**SCP**				df = 1		df = 1			
						A = 0.171[2], 0.162[5], 0.161[10]		A = 0.119[2], 0.088[5], 0.082[10]			
**(C)**											
**Polycultures**			**Neutral**		**Production**			**Welfare**			
			Weighting coefficients								
			1		2	5	10	2	5		10
**SC**	*rs*		15		14	12	10	15	16		16
	rank		2nd		2nd	2nd	1st	2nd	2nd		3rd
**SCP**	*rs*		10		10	10	11	10	10		11
	rank		1st		1st	1st	2nd	1st	1st		1st
**SM**	*rs*		17		18	20	21	16	16		15
	rank		3rd		3rd	3rd	3rd	3rd	2nd		2nd

**Table 5 life-12-01315-t005:** (A) Global results of multivariate permutation tests and (B) ranking results. The candidate polycultures corresponded to CRP (goldfish, common roach, European perch) and CRG (goldfish, common roach, ruffe). *rs* corresponds to the sum of the ranks. The numbers in brackets corresponds to the level of weighting. Significance = *p* < 0.05.

**(A)**											
**Priorities**		**Weighting Coefficients**									
		1		2			5			10	
Neutral-weighting		df = 1		-			-			-	
		A = 0.103									
Production-weighting		-		df = 1			df = 1			df = 1	
				A = 0.104			A = 0.104			A = 0.105	
Welfare-weighting		-		df = 1			df = 1			df = 1	
				A = 0.094			A = 0.091			A = 0.091	
**(B)**											
**Polycultures**		**Neutral**	**Production**					**Welfare**			
		Weighting coefficients									
		1	2		5	10		2	5		10
**CRG**	*rs*	17	18		21	22		17	17		17
	rank	1st	1st		1st	1st		1st	1st		1st
**CRP**	*rs*	34	33		30	29		34	34		34
	rank	2nd	2nd		2nd	2nd		2nd	2nd		2nd

**Table 6 life-12-01315-t006:** (A) Global, (B) Pair-wise results of multivariate permutation tests and (C) ranking results. The candidate polycultures correspond to PS (pikeperch, sterlet), PT (pikeperch, tench), and PST (pikeperch, sterlet, tench). *rs* corresponds to the sum of the ranks. The numbers in brackets correspond to the level of weighting. Significance = *p* < 0.05.

**(A)**										
**Priorities**		**Weighting Coefficients**								
		1	2			5		10		
Neutral-weighting		df = 2	-			-		-		
		A = 0.148								
Production-weighting		-	df = 2			df = 2		df = 2		
			A = 0.139			A = 0.136		A = 0.135		
Welfare-weighting		-	df = 2			df = 2		df = 2		
			A = 0.161			A = 0.170		A = 0.172		
**(B)**										
**Priorities**	**Polycultures**	**PS**				**PT**				
Neutral-weighting	**PT**	df = 1								
		A = 0.871								
	**PST**	df = 1				df = 1				
		A = 0.072				A = 0.196				
Production-weighting	**PT**	df = 1								
		A = 0.099[2], 0.103[5], 0.103[10]								
	**PST**	df = 1				df = 1				
		A = 0.050[2], 0.040[5], 0.039[10],				A = 0.191[2], 0.191[5], 0.192[10]				
Welfare-weighting	**PT**	df = 1								
		A = 0.079[2], 0.074[5], 0.074[10]								
	**PST**	df = 1				df = 1				
		A = 0.104[2], 0.121[5], 0.124[10]				A = 0.104[2], 0.121[5], 0.124[10]				
**(C)**										
**Polycultures**		**Neutral**		**Production**			**Welfare**			
			Weighting coefficients							
			1	2	5	10	2		5	10
**PS**		*rs*	12	13	14	15	13		13	13
		rank	1st	2nd	2nd	2nd	1st		2nd	2nd
**PST**		*rs*	13	11	10	10	15		18	19
		rank	2nd	1st	1st	1st	3rd		3rd	3rd
**PT**		*rs*	17	18	18	18	14		11	10
		rank	3rd	3rd	3rd	3rd	2nd		1st	1st

## Data Availability

The detailed trial protocols of each test case and all datasets, as well as R-script for polyculture assessment, are available on FigShare (see [49,50,51,63]).

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
