# Peer review of "Finding the Best Match: A Ranking Procedure of Fish Species Combinations for Polyculture Development"

_life, 2022, doi:10.3390/life12091315_

Round 1

Reviewer 1 Report

Insert economic aspects of the species.

Author Response

Reviewer 1:

Reviewer 1’s comment: Insert economic aspects of the species.

Authors’ response: Modifications have been done. The economic aspects of the species have been inserted in the revised version of the manuscript (see section 2.2.).

Reviewer 2 Report

Date: 03/08/2022

Ref. No. LIFE

Title: Finding the best match: a ranking procedure of fish species 2 combinations for polyculture development.

Dear Editor,

Manuscript is well structured and results are very well presented.

Please find comments:

1.      Abstract is written poorly. Aim and objective is clear but it does not have clear methodology, measurable result (data) and conclusion. Different test class must have been given and their result must have been discussed here.

2.      What was biomass in each class? Is it kept same?

3.      Discussion section is lengthy, which may be reduced.

Thanks,

Regards

Author Response

Reviewer 2:

Reviewer 2’s comment 1: Abstract is written poorly. Aim and objective is clear but it does not have clear methodology, measurable result (data) and conclusion. Different test class must have been given and their result must have been discussed here.

Authors’ response: The modifications have been done in the revised version of the manuscript (see Abstract section) in order to integrate methodology, main results, and conclusions. We did not provide the specific results of each test case in order to respect the word number limit of the abstract. Moreover, since our study aimed at highlighted potential of our procedure rather than the specific best polyculture(s) for pikeperch or goldfish among those analyzed, we preferred focusing on results reflecting the relevance of our procedure.

Reviewer 2’s comment 2: What was biomass in each class? Is it kept same?

Authors’ response: This information is provided in supplementary materials because of the word number restriction in Life. According to the journal guidelines, we uploaded the supplementary materials on FigShare (see DOIs in reference list, number [49–51,63]). We also provided the specific answer to your question below:

At the beginning of the trial:

  • Test case 1

A biomass of 4 kg m-3 was placed in each experimental unit (2 m3), with a distribution of

  • 100% pikeperch
  • 50% pikeperch - 50% common carp
  • 50% pikeperch - 50% black-bass
  • 33% pikeperch - 33% common carp - 33% European-perch
  • Test case 2

A biomass of 3 kg m-3 was placed in each experimental unit (0.3 m3), with a distribution of

  • 100% goldfish
  • 33% goldfish – 33% roach – 33% European-perch
  • 33% goldfish – 33% roach – 33% ruffe
  • Test case 3

We used a fixed number of 36 fishes per experimental unit (0.3 m3), with a distribution of:

  • 36 pikeperch
  • 18 pikeperch - 18 tench
  • 18 pikeperch - 18 sterlet
  • 12 pikeperch - 12 tench - 12 sterlet

Reviewer 2’s comment 3: Discussion section is lengthy, which may be reduced.

Authors’ response: We have shortened the discussion a bit. Nevertheless, the length of the discussion is a little less than 1.5 page to discuss the results, limitations and possible future developments of our polyculture classification procedure. This seems to us already concise enough.

Reviewer 3 Report

Amoussou and collaborators proposed an interesting method for improving animal welfare and the quality of products in aquaculture (identification number: life-1852576-peer-review-v1). The authors compared a monoculture vs polyculture by considering several physiological and behavioural fish traits and stakeholder priorities with a multivariate approach. Results showed i) the hypothesized stockholder priorities effectively affect the quality/efficacy of polycultures ii) multi-trait assessment provides more resilient results for assessing the status of fish in aquacultures.

The paper is professionally written. I really enjoyed the discussion on the limitations of the ranking procedure used, however, the authors well presented their interesting data and supported from previous literature. I consider the work by Amoussou and colleagues suitable for being considered for publication. I have some minor methodological comments that the authors would be considered for improving their study.

Authors reported detailed information as supplementary materials. I was unable to find any attached supplements.

Lines 108-118. The first step: The biological trait selection.

- the individuals’ final weight is influenced by their initial condition before the start of the rearing period. Why did you consider only the final weight of fish? If initial weight were available information, the growth rate would be another potential trait to be considered of.

- How did you calculate the survival rate?

- The period in which the Fulton condition index (FCF) was measured is missing from the text.

Lines 120-127. Behavioral and physiological traits

- I would ask whether positive aspects of rearing multiple species could be considered in the analysis, i.e., species without physiological and behavioural overall that could obtain advantages in living together.

Lines 147-160. The third step: weighting the traits

- I suggest reporting some references that would help the reader to understand and verify the robustness of the adopted procedure. In case there were no earlier studies in which the proposed weights have been used, their choice might be motivated and validated.

Lines 181-187 and Table 3. Test cases

- I would like to ask the authors whether my interpretation is correct. Table 3 shows the polyculture tanks in which two or three species were maintained. The highlighted names are the fish species maintained in the monoculture tanks. Therefore, your interest has mainly focused on verifying whether polyculture conditions improve the income and animal welfare of Pikeperch and Goldfish compared to monoculture tanks. Why did you choose these two species? Is it possible that other species suffer more from the polyculture conditions? Because two or more species are maintained together, I expect all fish species to be considered in the overall analysis.

- The conditions in which fish were maintained, i.e., fish density and abiotic factors, between polyculture and monoculture should be reported.

- Latin name of European-perch is missing

- How many replicas have you had for each mono/polyculture tank?

- Finally, the Goldfish groups were kept with three species. Are there any reasons for this choice?

Author Response

Reviewer 3:

Amoussou and collaborators proposed an interesting method for improving animal welfare and the quality of products in aquaculture (identification number: life-1852576-peer-review-v1). The authors compared a monoculture vs polyculture by considering several physiological and behavioural fish traits and stakeholder priorities with a multivariate approach. Results showed i) the hypothesized stockholder priorities effectively affect the quality/efficacy of polycultures ii) multi-trait assessment provides more resilient results for assessing the status of fish in aquacultures.

The paper is professionally written. I really enjoyed the discussion on the limitations of the ranking procedure used, however, the authors well presented their interesting data and supported from previous literature. I consider the work by Amoussou and colleagues suitable for being considered for publication. I have some minor methodological comments that the authors would be considered for improving their study.

Reviewer 3’s comment 1: Authors reported detailed information as supplementary materials. I was unable to find any attached supplements.

Authors’ response: According to the guidelines of Life, all the supplementary materials have been uploaded on FigShare. Links to access supplementary materials are provided in the reference list of the manuscript (DOIs in reference list, number [49–51,63]).

Lines 108-118. The first step: The biological trait selection.

Reviewer 3’s comment 2: The individuals’ final weight is influenced by their initial condition before the start of the rearing period. Why did you consider only the final weight of fish? If initial weight were available information, the growth rate would be another potential trait to be considered of.

Authors’ response: We think there is a potential misunderstanding because the growth rate was included in our analyses. For instance, at the beginning of the procedure (step 1), for the category of morphological traits, we had considered the final weight (Wf) but also the specific growth rate (SGR) (see section 2.1.1.). In addition, although in the rest of the procedure (step 2) correlations were detected between the SGR and some traits (Wf: R2 = 1, p-value < 0.01, test case 1 and (Fulton condition index: R2 = 1, p-value < 0.01, test case 2), the SGR was retained (see sections 3.1. and 3.2.).

Reviewer 3’s comment 3: How did you calculate the survival rate?

Authors’ response: The survival rate (%) corresponded to the ratio of initial to final number of fish. We multiplied the ratio by 100. This information is available in detailed protocol on FigShare; see reference number [49].

Reviewer 3’s comment 4: The period in which the Fulton condition index (FCF) was measured is missing from the text.

Authors’ response: The FCF was calculated after a trial periods of 90 days (test cases 1 and 2) and 60 days (test case 3). Please find the information in the manuscript (section 2.2., table 3) and all details about trait measurements in detailed protocol on FigShare (see DOIs in reference list, number [49]).

Lines 120-127. Behavioral and physiological traits

Reviewer 3’s comment 5: I would ask whether positive aspects of rearing multiple species could be considered in the analysis, i.e., species without physiological and behavioural overall that could obtain advantages in living together.

Authors’ response: We may not have understood your comment, but our procedure already allows detecting the benefits, from a production and welfare point of view, of a polyculture compared to a monoculture. If it is still necessary after modifications to the manuscript, could you please clarify your question?

Lines 147-160. The third step: weighting the traits

Reviewer 3’s comment 6: I suggest reporting some references that would help the reader to understand and verify the robustness of the adopted procedure. In case there were no earlier studies in which the proposed weights have been used, their choice might be motivated and validated.

Authors’ response: There are currently no reference in the literature that provide weighting coefficients for production and welfare traits in a polyculture context. However, the aim of our study was to apply our procedure on test cases in order to propose a proof of concept to the readers. In this context, we arbitrarily defined weighting coefficients to show the influence of these on the results of our procedure. We emphasized this point in section 2.1.3. and we drew the attention of future users of the procedure to the imperative need to define these coefficients in collaboration with the stakeholders (see section 4.3.)

Lines 181-187 and Table 3. Test cases

Reviewer 3’s comment 7: I would like to ask the authors whether my interpretation is correct. Table 3 shows the polyculture tanks in which two or three species were maintained. The highlighted names are the fish species maintained in the monoculture tanks. Therefore, your interest has mainly focused on verifying whether polyculture conditions improve the income and animal welfare of pikeperch and goldfish compared to monoculture tanks.

  • Why did you choose these two species?
  • Is it possible that other species suffer more from the polyculture conditions?
  • Because two or more species are maintained together, I expect all fish species to be considered in the overall analysis.

Authors’ response: We have responded in the order of the points you raised.

  • These two species had been chosen based on their economic interest. As also request by the reviewer 1, we added rationalizations about the choice of these two species in the section 2.2..
  • You are right. We considered this potential problem in the discussion (section 4.3) and provided guidelines to future procedure users to avoid it.
  • You are right, it would be better to consider the consequences of polyculture for all the associated species (see DOI 1016/j.aquaculture.2022.738438). However, for our test cases, such an information was not available (i.e., all studied traits had not been measured for associated species, and monoculture of all associated species had not been performed), which did not make it possible to assess polyculture consequences for all species. We clearly underlined in the section 4.3 that a stronger polyculture assessment must consider the consequences for all combined species. We recommended to future users of our approach to proceed as such.

Reviewer 3’s comment 8: The conditions in which fish were maintained, i.e., fish density and abiotic factors, between polyculture and monoculture should be reported.

Authors’ response: The conditions in which fish were maintained in monoculture and polyculture were available in the supporting information uploaded on FigShare (see DOIs in reference list, number [49]). We also added information about the applied conditions in the table below:

Test cases

Modalities

Modalities

Rearing volume (m3)

Trial

duration (days)

Number of replicates

Number of EU

Fish density

Fish number

Abiotic factors

(average values)

1

·  Monoculture

·  Pikeperch (Sander lucioperca)

2

90

3

12

4 kg m-3

-

Water temperature: 20.9 ± 0.2 °C; dissolved oxygen: 7.4 ± 0.6 mg. L-1; pH: 7.5 ± 0.2; ammonia: 0.02 ± 0.05 mg. L-1; nitrite: 0.07 ± 0.18 mg. L-1); luminosity: 20 lx and photoperiod: 10 L: 14 D

·  Polycultures

·  Pikeperch, Common carp (Cyprinus carpio) [SC]

·  Pikeperch, Black-bass (Micropterus salmoides) [SM]

·  Pikeperch, Common carp, European-perch (Perca fluviatilis) [SCP]

2

·  Monoculture

·  Goldfish (Carassius auratus)

0.3

90

5

15

3 kg m-3

-

Water temperature: 21.12 ± 1.07 °C; dissolved oxygen: 7.34 ± 0.56 mg. L-1; pH: 7.42 ± 0.45; ammonia: 0.06 ± 0.06 mg. L-1; nitrites: 0.08 ± 0.06 mg. L-1; luminosity: 20 Lux and photoperiod: 10 L: 14 D

·  Polycultures

·  Goldfish, Roach (Rutilus rutilus), Ruffe (Gymnocephalus cernua) [CRG]

·  Goldfish, Roach, European-perch [CRP]

3

·  Monoculture

·  Pikeperch

0.3

60

3

12

-

36 fish/EU

Water temperature: 20.45 ± 0.75 °C; dissolved oxygen: 6.08 ± 0.92 mg. L-1; pH: 7.46 ± 0.25; ammonia: 0.17 ± 0.18 mg. L-1; nitrites: 0.11 ± 0.10 mg. L-1; luminosity: 20 Lux and photoperiod: 10 L: 14 D

·  Polycultures

·  Pikeperch, Tench (Tinca tinca) [PT]

·  Pikeperch, Sterlet (Acipenser ruthenus) [PS]

·  Pikeperch, Sterlet, Tench [PST]

Reviewer 3’s comment 9: Latin name of European-perch is missing

Authors’ response: Modification has been done.

Reviewer 3’s comment 10: How many replicas have you had for each mono/polyculture tank?

Authors’ response: This information was provided in the detailed protocol available on FigShare (see DOIs in reference list, number [49]). However, we also added it in the table 3 of the manuscript in the revised version.

Reviewer 3’s comment 11: Finally, the Goldfish groups were kept with three species. Are there any reasons for this choice?

Authors’ response: The aim of the study was to exemplify the use of our procedure. Therefore, the test case was based on species that are of economic interest and/or were available for polyculture trials in RAS. We used polycultures of two or three species, mainly for pragmatic reasons (difficulty in finding a large number of species at the right stage of development at the same time and available from our fish suppliers.), but we argue that they allow us to put our procedure to the test.

References

For references cited in our responses to reviewers, please refer to the reference list in the revised manuscript.
